# An Intervention of Four Weeks of Time-Restricted Eating (16/8) in Male Long-Distance Runners Does Not Affect Cardiometabolic Risk Factors

**DOI:** 10.3390/nu15040985

**Published:** 2023-02-16

**Authors:** Christine E. Richardson, Ashley P. Tovar, Brian A. Davis, Marta D. Van Loan, Nancy L. Keim, Gretchen A. Casazza

**Affiliations:** 1Department of Nutrition, University of California, Davis, CA 95616, USA; 2USDA, ARS, Western Human Nutrition Research Center, Davis, CA 95616, USA; 3Physical Medicine and Rehabilitation, University of California Davis Medical Center, Sacramento, CA 95816, USA; 4Department of Kinesiology, California State University, Sacramento, CA 95819, USA

**Keywords:** resting energy expenditure, insulin, glucose, blood pressure, body composition, fat mass, cholesterol, lipoproteins, intermittent fasting

## Abstract

Timing of nutrient intake for athletes may affect exercise performance and cardiometabolic factors. Our objective was to examine the effect of time-restricted eating (TRE) on cardiometabolic health. Using a cross-over study design, 15 endurance-trained male runners were randomized to either a normal dietary pattern (ND) first (12 h eating/fasting times) followed by time-restricted eating (TRE) pattern (16 h fast; 8 h eating) or the reverse, with a 4-week washout period between interventions. Body composition, resting energy expenditure, blood pressure and serum insulin, glucose and lipids were measured using standard laboratory methods. Exercise training and dietary intake (calories and macronutrients) were similar across interventions. No significant differences were observed in resting energy expenditure, markers of insulin resistance, serum lipids or blood pressure. Body composition did change significantly (*p* < 0.05) with whole body fat mass (−0.8 ± 1.3 kg with TRE vs. +0.1 ± 4.3 kg with ND), leg fat mass (−0.3 ± 0.5 kg with TRE vs. +0.1 ± 0.4 kg with ND), and percent body fat (−1.0 ± 1.5% with TRE vs. +0.1 ± 1.3% with ND) declining more in the TRE intervention, with no change in fat-free mass. This study is one of a few to investigate the effects of an isocaloric 16/8 TRE eating pattern in trained endurance athletes and confirms no change in cardiometabolic risk factors. In conclusion, TRE is not detrimental to cardiometabolic health in endurance-trained male runners but could be beneficial on exercise performance by reducing fat mass.

## 1. Introduction

There are various forms of intermittent fasting (IF). This manuscript will focus on time-restricted eating (TRE), a form of IF that is growing in popularity. This regimen involves a decrease in the window of eating each day, essentially extending the overnight fast, which may or may not result in a decrease in caloric intake. This dietary approach can be used by overweight/obese individuals looking to lose body fat and improve their overall health. Likewise, this diet can also be used by normal weight and otherwise healthy persons who want to improve their health independent of weight loss. Studies show that by extending the overnight fast, as done with TRE, one can expect to see a decrease in risk for various metabolic conditions such as cardiovascular disease, type 2 diabetes, and more. One iteration is the “16/8” diet, where individuals fast for 16 h of the day and eat ad libitum for the remaining 8 h [1,2,3,4,5]. Competitive endurance athletes believe they may improve their exercise performance from this type of dietary alteration, despite a dearth of scientific evidence to substantiate these claims [6,7]. To date, only a few studies in athletes have studied TRE, some focused primarily on resistance-trained athletes [8,9,10], and one other focused on runners [11]. Moro and associates investigated the effects of 8 weeks of an isocaloric, 16/8 diet in resistance-trained males and reported a significant decrease in fat mass, despite no change in fat-free mass and no significant changes in resting energy expenditure and blood total and LDL cholesterol. However, in the TRE group, blood HDL significantly increased by 7%, glucose decreased by 11%, insulin decreased by 36% and triglycerides decreased by 7% [8]. Tinsley et al. [9]. conducted a similar study with 8 weeks of an isocaloric, 16/8 diet in resistance-trained females and also found a significant decrease in body mass and fat mass with no change in fat-free mass. Brady et al. [11]. reported that middle- and long-distance runners experienced a 1.9 kg reduction in body mass after 8 weeks of TRE but had no change in body composition and no change in fasting blood glucose, insulin or triglyceride concentrations. However, none of these studies used a randomized crossover design, and the Brady et al. [11] study did not match calorie or macronutrient contents between interventions. In our companion paper, which examined the effects of 4 weeks of 16/8 time-restricted eating in endurance-trained male runners, we found a significant decrease in fat mass with no change in 10 km time trial running performance [12]. However, we did not include measures of cardiovascular health in that paper.

One important consideration unique to endurance-trained athletes is that they are at risk for being in a low energy availability (LEA) state, which can lead to cardiovascular dysfunction as evidenced by disturbances in the blood lipid profile [13,14] and orthostatic intolerance due to poor sympathetic circulatory responses [15,16]. Traditionally, LEA has been used to describe inadequate energy intake relative to energy expenditure. It is possible that adhering to a TRE dietary pattern could lead to similar negative health outcomes as it prolongs the state of which an athlete is in a state of negative energy balance and a small window for food intake can make it hard to reach the high caloric intake needed with high energy expenditures. Although endurance athletes are at a decreased risk for developing cardiovascular disease (CVD) through regular exercise [17] this doesn’t mean that this population is immune to this disease. Many athletes consume a highly processed, high-glycemic, low fiber diet that can cause negative physiological responses like an increase in reactive oxygen species production and pro-inflammatory mediators that may contribute to the progression of arterial streaks and atherosclerosis [18]. Studies have also shown that athletes can be at risk for calcification of arteries while being in a constant inflammatory state due to the body’s need to address the constant damage to bones, joints, and tendons during prolonged bouts of exercise such as running and cycling [19,20,21]. Lastly, studies have shown that many athletes adopt a sedentary lifestyle after sustaining an injury or after retiring from sport and will later display the same risk factors associated with metabolic syndrome and CVD such as disturbances in the lipid profile [22]. Consequently, endurance athletes should be considered when investigating dietary alteration effects such as TRE and the effects on CVD risk.

Peer-reviewed literature is sparse and contains conflicting information about the effects of TRE in athletes. In addition, we know of no study that had used a longitudinal randomized crossover design. Thus, the aim of this study is to examine the effects of four weeks of the 16/8 TRE diet compared to a normal eating window of 12 h fasting and 12 h eating on body composition, resting energy expenditure, and biomarkers of cardiometabolic disease risk using a randomized crossover study design in competitive male endurance runners.

## 2. Materials and Methods

### 2.1. Experimental Design

In this cross-over intervention, subjects were randomly assigned to start the study with either a traditional 12-h eating window (12/12) (ND) or a time-restricted 8-h eating window (16/8) (TRE). Diets were self-selected, and subjects were instructed to consume isocaloric diets of the similar macronutrient composition based on diet records kept during the first arm of the study. A normal eating window of 12 h fasting and 12 h eating was modeled after the study designed my Moro, et al. to compare results in resistance-trained versus endurance-trained athletes. Details of the study design, subjects and performance testing are provided in Tovar, et al. [12]. Briefly, subjects chose their eating windows and were instructed to consume all caloric intake within the same 12 h or 8 h period during their assigned diet intervention. Fasting periods outside of the feeding windows only allowed for water and non-caloric beverages, such as unsweetened black coffee or plain tea. Subjects were also asked to maintain the same training regime for both arms of the study, which was monitored using polar activity watches. After the first 4-week intervention, a 2-week washout period was scheduled, followed by the other 4-week intervention. A 2-week washout period was the original intent, but a washout period of 3 or 4 weeks was used for some subjects to accommodate injury, sickness or subject schedule. Each subject performed their exercise training in a fasted or fed state based on their preferred habits for both arms of the study. Subjects were instructed to maintain their normal eating patterns prior to starting the study and while during the washout period. Subjects completed a 3-day dietary log prior to the first test day to provide baseline dietary data. Subjects also completed a 3-day dietary log during the washout period to ensure they resumed their normal eating patterns. Prior to starting the first intervention arm, subjects completed a familiarization visit for consenting, reviewing the design, procedures and expectations of the study; obtaining medical clearance for performance testing; and training for keeping diet records. Participants provided written informed consent approved by the Institutional Review Board at the University of California at Davis, IRB protocol #1223350. The protocol was listed as identifier NCT03569852 by the national clinical trials public website.

### 2.2. Subjects

To be included, subjects were required to be born male, have been actively training for the past 3 years, run ≥ 32 km/wk, have competed in a race ≥ 5 km within the last 12 mo, have a maximal oxygen consumption (VO_2_ max) ≥ 40 mL/kg/min and be weight-stable for the past 6 mo. Subjects were excluded if they were currently taking prescription medications or dietary supplements that caused any metabolic alterations, had any injuries that limited their ability to train or perform the exercise testing regimen, were smokers or had been consistently smoking during the last 6 mo or were adhering to any form of a restrictive diet as defined by a >20% deviation from the suggested macronutrient ranges for endurance athletes by the American College of Sports Medicine [23]. Reported chronic diseases or serious health concerns as determined by the study physician disqualified them from participation.

### 2.3. Familiarization Visit

Following recruitment, subjects visited the USDA, ARS, Western Human Nutrition Research Center (WHNRC) at the University of California at Davis campus to learn the study parameters and receive instruction using in-house equipment. After signing the informed consent, subjects then completed a health history questionnaire prior to collecting height and weight. After receiving medical clearance by a study physician, subjects engaged in a self-selected 10-min warm up on a treadmill (TMX425 medical treadmill, Trackmaster, Newton, KS, USA), and then completed a treadmill graded test to determine VO_2_peak. A metabolic cart (TrueOne 2400, ParvoMedics, Sandy, UT, USA) was used for collecting respiratory gases and calculating gas exchange variables. The cart was calibrated before each test (flow rates 50–400 l*min^−1^ and at room air and with a standard gas mixture of 16% O_2_ and 4% CO_2_). A watch (5410, Polar, Woodbury, NY, USA) was used to monitor heart rate. Subjects’ starting running speed was determined based on an estimated training pace for a 5 km run given by each participate in order to maintain a test duration of 12–15 min. Speed increased by 0.8 kmph every 2 min with a constant 1% grade. Rate of perceived exertion (RPE) was asked every 2 min using a 10-point scale (12). Criteria for a maximal test included at least 2 of the following, a plateau in VO_2_peak with increasing speed, a maximal heart rate exceeding 90% of predicted (220-age), the rate of perceived exertion > 9 and a respiratory exchange ratio (RER) > 1.10. After completing the test, study investigators instructed subjects on proper and accurate food documentation and how to maintain consistent caloric and macronutrient intake for both arms of the study. They were asked to complete a 3-day diet record to assess dietary habits. If any individual’s diet log showed evidence of any type of dietary restriction, they were excluded from the study.

### 2.4. Test Day Protocol

Four research test days were scheduled with one at the beginning and end of each of the 4-week periods. Subjects arrived following an overnight fast and were asked to only drink water. They had been instructed to refrain from exercise for 24 h, follow a consistent hydration pattern, and consume the same meal the night before each test day. Upon arrival, body mass was measured on a calibrated electronic scale (Tanita BWB-627A Class III electronic scale; Toledo Scale), height was measured with a wall-mounted stadiometer (Ayrton Stadiometer, Model S100; Ayrton Corp., Prior Lake, MN, USA) and resting energy expenditure (REE) was measured using a metabolic cart (TrueOne 2400, ParvoMedics, Sandy, UT, USA).

### 2.5. Resting Energy Expenditure

Subjects rested for 10 min in a supine position in a dark, quiet room. Respiratory gases were then collected for 20 min. Data from the first 5 min were excluded from analysis to account for subject adjustment to the protocol. The Weir equation was used to determine rate of kcal/day (Equation (1)). Because nitrogen excretion is minimal in such a short period of time, the nitrogen correction was ignored [24].
Kcal/day = (3.94 × VO_2_) + (1.11 × VCO_2_)(1)

Energy derived from total carbohydrate and lipid oxidation was calculated using the following equations (Equation (2)) assuming protein oxidation was close to zero for this brief time period [25].
% energy from carbohydrate = (4.55 × VCO_2_) − (3.21 × VO_2_) × 100% energy from lipid = (1.67 × VO_2_) − (1.67 × VCO_2_) × 100(2)

### 2.6. Blood Pressure

Immediately after the REE determination and before standing, resting blood pressure (BP) was measured manually with a single-hosed sphygmomanometer and stethoscope by the same investigator for all trials. After supine blood pressure was measured, subjects were asked to stand for 5 min before collecting standing blood pressure to determine orthostatic tolerance [16].

### 2.7. Body Composition

A whole-body dual-energy X-ray absorptiometry (DXA) (Hologic Discovery QDR Series 94994; Hologic, Inc.) scan was performed for determination of body composition. The DXA scanner was calibrated prior to each use by the same trained and licensed technician. The scan provided values for total fat mass, total lean mass, body fat percentage, peripheral lean mass, peripheral fat mass, an estimate of android and gynoid fat mass distribution, bone density, bone density z-score and bone mineral mass. All DXA scans were analyzed by a single operator to minimize variance in the results.

### 2.8. Blood Analyses

Fasting blood samples were collected using sterile, disposable materials by a licensed phlebotomist. Blood was drawn directly into SST vacutainers. SST tubes sat at room temperature for 30 min and were then centrifuged in a refrigerated Centra CL3R (International Equipment Co.) for 10 min at 100× *g* at 10 °C. Next, 50 uL of serum was inserted into a basic metabolic reagent disk and placed into a Piccolo Xpress Chemistry Analyzer (Abbott, Princeton, NJ, USA) for determination of glucose and CVD risk markers, which included the following: glucose, total cholesterol, HDL, LDL, VLDL, non-HDLc and triglycerides. Ratios of total cholesterol to HDL and LDL to HDL were calculated.

Insulin was measured in duplicate using Meso Scale Delivery (MSD) Multi-plex Assay System and were conducted according to the manufacturer’s instructions. Briefly, 150 uL of Blocker A was added to each well of the MSD plate, which was then sealed, incubated and shook (1000 rpm) for one hour at room temperature. The plate was then washed with phosphate-buffered saline plus 0.05% Tween-20 (PBS-T), and 50 uL of the sample and standard were added to each well. The plate was then sealed, incubated and shook (1000 rpm) for 2 h at room temperature. The plate was washed again with PBS-T and then 25 uL of detection antibody solution was added to each well. The plate was then sealed, incubated and shook (1000 rpm) at room temperature for one hour. The plate was washed for a final time with PBS-T and then 150 uL of Read Buffer T was added to each well. The plate was read on the MSD QuickPlex SQ 120 imager and quantified using an 8-point standard curve. Insulin concentrations were used to calculate HOMA-IR and QUICKI (Equation (3)).
HOMA-IR = (Insulin (µU/mL) × glucose (mg/dL)) ÷ 405QUICKI = 1 ÷ (Log insulin (µU/mL) + log glucose (mg/dL))(3)

### 2.9. Statistical Analysis

All analyses were done using JMP Pro 14 (SAS Institute Inc, Cary, NC, USA). For each variable, the change from pre- to post-intervention was calculated, and the effects of diet intervention, sequence and interaction were analyzed using a mixed linear model. All variables that were not considered normally distributed were normalized using a Johnson transformation. Post hoc analysis was performed with the Tukey’s test. Significance was set at *p* ≤ 0.05. All values are presented as means ± SD unless indicated otherwise. Details on power analysis are described in Tovar et al. [19]. Briefly, power analysis determined a total of 16 subjects were needed to achieve 80% power with an alpha error probability of 0.05.

## 3. Results

### 3.1. Subject Enrollment and Retention

Healthy, endurance-trained male runners, n = 27, provided consent. Three subjects were disqualified after consent due to cardiovascular abnormalities or not meeting the VO_2_max criteria. Four subjects dropped out after the familiarization visit due personal commitments. Five subjects did not complete both arms of the study due to personal commitments (n = 4) or early cessation of data collection related to COVID-19 restrictions (n =1). Baseline characteristics of the final completers (n=15) can be found in Table 1.

Because subjects were given the liberty to choose their window of eating, there was some variability. For the TRE arm, the average time to begin eating was 11.1 ± 1.95 h. Likewise, during the normal diet, most subjects began eating at 7.8 ± 1.4 h.

Table 2 reflects dietary intake from the 3-day food log done weekly consisting of one high intensity exercise day, one medium intensity exercise day and one rest day. No significant differences were found between groups.

### 3.2. Resting Energy Expenditure

After 4 weeks, there was no significant difference between the TRE and ND interventions for resting energy expenditure, resting respiratory exchange ratio, resting energy expenditure by body mass and resting energy expenditure by fat-free mass (Table 3).

### 3.3. Body Mass and Composition

We have previously reported [12] that fat mass significantly decreased by 7% and body fat percentage by 6% in the TRE group compared to no change in the ND group. These changes along with segmental body composition data are reported in (Table 4). Leg fat mass decreased significantly (*p* = 0.03) in the TRE group, but there were no significant differences for arm fat mass, arm fat-free mass, leg fat-free mass, trunk fat mass, trunk fat-free mass or the android/gynoid ratio between interventions. There was a sequence effect for fat mass, body fat percentage and leg fat mass for subjects randomized to the TRE diet first.

### 3.4. Bone Mineral Density

DXA results for bone mineral density and bone mineral density z-score can be found in Table 5. No significant differences were found between diet interventions.

### 3.5. Insulin Resistance and Sensitivity

Fasting blood glucose and insulin concentrations were not different between interventions and are shown in Table 6. Calculated HOMA-IR, a marker of insulin resistance, and QUICKI, a marker of insulin sensitivity, were not different between interventions and are shown in Table 6.

### 3.6. Blood Pressure

No significant differences were observed for systolic or diastolic blood pressure in the supine position or after 5 min of standing between diet interventions (Table 7).

### 3.7. Circulating Lipids and Lipoproteins

No differences in total cholesterol and triglycerides, as well as VLDL-, LDL-, HDL- or non-HDL-cholesterol (nHDLc), were detected between diet interventions (Table 8).

## 4. Discussion

This report extends the original findings from our TRE study in endurance-trained athletes [12] in which we observed that the TRE intervention resulted in the loss of total fat mass and body fat percentage with no change in 10 km time trial performance. In the present report, except for discovering a significant decrease in the change in segmental leg fat mass with TRE compared to normal diet, no other changes in segmental body composition, bone mineral content, resting energy expenditure, resting respiratory exchange ratio or cardiometabolic disease markers were found between the TRE and normal dietary interventions.

The present study is one of just a few studies to investigate the effects of the 16/8 time-restricted eating diet in trained athletes. To our knowledge, there is only one other study conducted in endurance-trained athletes. Brady et al. [11] studied the 16/8 diet in runners who were randomly assigned to either the TRE intervention or a control diet for 8 weeks. In contrast to our study, calorie and macronutrient contents were not matched between dietary interventions. They reported that TRE resulted in a reduction in body mass, but not fat mass or fat-free mass, and a reduction in energy intake of ~250 kcal/d was recorded mid-study. The decreased energy intake could explain why Brady et al. found a decrease in body mass, whereas our study did not. As seen with our study, health-related cardiometabolic factors—fasting glucose, insulin and triglyceride concentrations—were not affected by TRE.

Moro et al. studied the 16/8 diet in resistance-trained males while maintaining similar caloric and macronutrient intake and in a similar design, Tinsley et al. investigated a 16/8 TRE diet combined with an 8-week resistance training program, but in resistance-trained female athletes [8,9]. These results are similar to our study, where we found no change in fat-free mass but found decreased whole body and leg fat-free mass with a 16/8 TRE diet matched for calories and macronutrient content.

Interestingly, considering that the energy and macronutrient intake and running distance each week were not significantly different between interventions, a significant decrease in RER was not observed, indicating no significant shift to utilize more fat and less carbohydrate and/or glycogen. There was a slight observed decrease in RER with TRE, and it is conceivable that RER would have decreased significantly if the study duration was longer than 4 weeks or if the present study employed more subjects. Future studies should include a longer intervention to determine if dietary duration may affect this result.

It should be noted that it does appear that maintaining energy intake is vital when adopting TRE to achieve goals related to loss of fat mass. The 650 kcal deficit reported in the Tinsley et al. study could cause the body to enter a “conservation mode” or “survival mode” where the body perceives starvation and will employ a number of metabolic adaptations, such as decreasing resting metabolic rate and improving metabolic efficiency, in an attempt to preserve energy and body fat [26,27]. The body may enter this state if the caloric content decreases too much while on a TRE diet; however, additional studies are needed to substantiate this.

The primary objective of our study was to determine if the TRE diet intervention affected cardiometabolic variables differently than the response to the normal diet intervention. Traditional clinical markers of cardiovascular disease were not affected by TRE in the present study. This is not surprising because they were actively training, and longer bouts of aerobic exercise lead to greater reliance on free fatty acids for energy production [28,29]. These results were similar to results reported by Brady, et al. [11], who also studied long-distance runners. In contrast, in resistance-trained males, 8 weeks of the 16/8 TRE diet resulted in increases in HDL cholesterol and decreases in blood glucose, insulin and triglycerides. Neither resting energy expenditure nor fat-free mass differed between diets, presenting findings consistent with Moro, et al. [8]. This suggests that the 16/8 diet may be a superior strategy for fat loss compared to continuous energy restriction, which is often associated with a decrease in fat-free mass and a concurrent decrease in resting energy expenditure [30,31].

There is growing evidence to support the notion that TRE can be utilized to combat insulin resistance; however, this has primarily only been studied in overweight and obese populations [32]. The present study suggests that adhering to TRE for 4 weeks does not significantly affect insulin resistance, insulin sensitivity or glucose metabolism. Given the fact that aerobic exercise is known to cause enhanced insulin sensitivity through similar physiological mechanisms as fasting, it is possible that there is little room for improvement with these specific mechanisms.

Several limitations to the present study should be discussed. For example, participants were allowed to choose the time frame for their window of eating. The window of eating was not standardized because a study by Hutchison et al. showed no difference in fasting glucose, insulin, triglycerides, non-esterified fatty acids or gastrointestinal hormones between early TRE (eating window 8 am to 5 pm) and delayed TRE (12 pm to 9 pm). While they did not examine a full lipid panel, this study provides some insight to the notion that the window of meal timing may not matter with TRE diets [33]. Additionally, controlling time of eating may have caused shifts in the subjects’ sleep/wake cycle, thereby altering hormones related to the circadian rhythm and introducing a new variable to the study design [34,35]. Furthermore, it was thought that allowing subjects to choose their window of eating may decrease the drop-out rate. Another limitation was self-reported diet recording. Although subjects were trained on proper portion sizes, etc., it is possible that energy or macronutrient intake may have been misreported as this is common in human studies. Additionally, meal timing with respect to exercise may have influenced the results. Although subjects were instructed to maintain their normal workout routine, it is possible that they altered the time of day they ran, which may have led to more fasted or fed workouts, thereby altering the results. Another potential limitation to this study is the duration of the washout, which was planned to be four weeks but fluctuated between subjects because of availability (mean 2.1 ± 0.7 weeks). Finally, the present study does have a small subject number at 15 when our power analysis showed a need for 16 subjects. COVID-19 restrictions were implemented before we could complete 16 subjects and thus our power only reached 76%, which introduces low statistical power for each outcome variable. It is conceivable that a larger sample size would have revealed differences between interventions.

## 5. Conclusions

These results suggest that endurance athletes adhering to an isocaloric 16/8 TRE dietary pattern for 4 weeks experienced no identifiable adverse changes in the cardiometabolic risk factors. Athletes who are trying to reduce fat mass before an event should consider adopting a TRE dietary pattern as opposed to a caloric deficit diet in order to maintain fat-free mass and resting energy expenditure.

## Figures and Tables

**Table 1 nutrients-15-00985-t001:** Baseline Characteristics ^1^.

Age (year)	28. 7 ± 5.2	Fat Free Mass (kg)	57.6 ± 7.6
Body Mass (kg)	73.5 ± 8.6	VO_2_max (mL/kg/min)	55.2 ± 6.7
Height (cm)	177.7 ± 6.6	Km run per week	52.9 ± 10.8
Body fat (%)	16.0 ± 5.6	Years training	7.8 ± 6.0
Fat Mass (kg)	12.0 ± 4.5		

^1^ Values are mean ± SD for 15 subjects. VO_2_, maximal oxygen consumption.

**Table 2 nutrients-15-00985-t002:** Dietary Intake ^1^.

	Normal Diet (12/12)	Time-Restricted Eating (16/8)	*p*-Value
Kcal/day	2513 ± 367	2421 ± 478	0.41
Carbohydrate (g/day)	284.8 ± 79.3	269.4 ± 68.4	0.27
Protein (g/day)	112.5 ± 27.1	113.1 ± 24.4	0.42
Fat (g/day)	97.5 ± 24.5	96.8 ± 33.0	0.91

^1^ Values are mean ± SD for 15 subjects. ND, normal Diet; TRE Time Restricted Eating.

**Table 3 nutrients-15-00985-t003:** Resting Energy Expenditure ^1^.

	Normal Diet (12/12)			Time-Restricted Eating (16/8)			
	Pre-	Post-	Change	Pre-	Post-	Change	*p* Value
REE (kcal)	1644 ± 361	1724 ± 277	80 ± 322	1689 ± 304	1698 ± 269	8 ± 158	0.33
REE/body mass (kcal/kg)	22.5 ± 4.1	23.7± 3.7	1.1 ± 4.3	23.0 ± 3.7	23.4 ± 3.4	0.4 ± 2.4	0.44
REE/FFM (kcal/kg)	28.6 ± 4.8	29.8 ± 4.6	1.2 ± 5.8	29.4 ± 4.0	29.5 ± 3.7	0.1 ± 3.0	0.40
RER	0.85 ± 0.04	0.85 ± 0.05	0.0 ± 0.07	0.86 ± 0.07	0.82 ± 0.05	−0.03 ± 0.08	0.19

^1^ Values are mean ± SD for 15 subjects. Abbreviations: resting energy expenditure (REE), respiratory exchange ratio (RER), fat-free mass (FFM). *p* values represent difference in the change values between diet interventions.

**Table 4 nutrients-15-00985-t004:** Body Weight and Composition ^1^.

	Normal Diet (12/12)			Time-Restricted Eating (16/8)			
	Pre-	Post-	Change	Pre-	Post-	Change	*p* Value
Body mass (kg)	73.0 ± 8.6	73.3 ± 8.7	0.4 ± 1.1	73.8 ± 8.6	73.0 ± 9.0	−0.8 ± 1.9	0.09
Whole body FM	11.7 ± 4.8	11. 8 ± 4.3	0.1 ± 4.3	12.3 ± 4.3	11.5 ± 4.4	−0.8 ± 1.3	* 0.05
Whole body FFM	57.6 ± 7.2	58.3 ± 7.8	0.8 ± 2.4	57.7 ± 7.3	57.8 ± 7.2	0.2 ± 1.7	0.47
Total leg FM (kg)	4.0 ± 1.6	4.1 ± 1.5	0.1 ± 0.4	4.2 ± 1.4	3.9 ± 1.4	−0.3 ± 0.5	* 0.03
Total leg FFM (kg)	9.4 ± 1.2	9.5 ± 1.3	0.1 ± 0.4	9.3 ± 1.3	9.4 ± 1.3	−0.1 ± 0.4	0.24
Trunk FM (kg)	5.7 ± 2.7	5.7 ± 2.6	−0.4 ± 1.5	6.0 ± 2.6	5.1 ± 3.0	−0.8 ± 1.8	0.18
Trunk FFM (kg)	28.3 ± 3.9	28.6 ± 3.9	0.1 ± 0.7	28.5 ± 4	28.6 ± 4.2	−0.2 ± 1.0	0.25
Total arm FM (kg)	1.1 ± 0.5	1.1 ± 0.5	0.0 ± 0.1	1.2 ± 0.5	1.1 ± 0.5	−0.1 ± 0.1	0.08
Total arm FFM (kg)	6.4 ± 0.9	6.3 ± 0.9	0.1 ± 0.2	6.4 ± 0.9	6.5 ± 0.9	−0.0 ± 0.5	0.67
Body Fat %	16.1 ± 5.7	16.2 ± 5.3	0.1 ± 1.3	16.8 ± 5.5	15.8 ± 5.2	−1.0 ± 1.5	* 0.04
Android/gynoid ratio	0.9 ± 0.2	0.8 ± 0.2	−0.0 ± 0.1	0.9 ± 0.2	0.8 ± 0.2	−0.0 ± 0.1	0.65

^1^ Values are mean ± SD for 15 subjects. Abbreviations: Fat mass (FM), fat-free mass (FFM), * significant difference in the change values between interventions. *p* values represent difference in the change values between diet interventions.

**Table 5 nutrients-15-00985-t005:** Bone Mineral Density ^1^.

	Normal Diet (12/12)			Time-Restricted Eating (16/8)			
	Pre-	Post-	Change	Pre-	Post-	Change	*p* Value
BMD	1.2 ± 0.1	1.2 ± 0.1	-0.0 ± 0.0	1.2 ± 0.1	1.2 ± 0.1	0.0 ± 0.0	0.32
BMD z-score	0.4 ± 1.0	0.5 ± 0.9	0.1 ± 0.7	0.5 ± 0.8	0.6 ± 0.8	0.1 ± 0.2	0.65

^1^ Values are mean ± SD for 15 subjects. Abbreviation: Bone mineral density (BMD). *p* values represent difference in the change values between diet interventions.

**Table 6 nutrients-15-00985-t006:** Markers of Insulin Resistance and Sensitivity ^1^.

	Normal Diet (12/12)			Time-Restricted Eating (16/8)			
	Pre-	Post-	Change	Pre-	Post-	Change	*p* Value
Glucose (mg/dL)	98.5 ± 6.3	95.8 ± 6.8	−2.7 ± 6.9	95.5 ± 6	94.6 ± 7.6	−0.9 ± 7.6	0.78
Insulin (μIU/mL)	6.7 ± 2.8	5.9 ± 2.9	−0.7 ± 3.1	6.6 ± 2.6	7.3 ± 3.5	0.7 ± 2.6	0.21
HOMA-IR	1.6 ± 0.7	1.4 ± 0.7	−0.2 ± 0.8	1.6 ± 0.6	1.7 ± 0.9	0.2 ± 0.6	0.16
QUICKI	0.36 ± 0.03	0.37 ± 0.03	0.01 ± 0.03	0.36 ± 0.02	0.36 ± 0.02	0.00 ± 0.03	0.27

^1^ Values are mean ± SD for 15 subjects. Abbreviations: Homeostatic model assessment for insulin resistance, HOMA-IR; quantitative insulin-sensitivity check index, QUICKI. *p* values represent difference in the change values between diet patterns.

**Table 7 nutrients-15-00985-t007:** Resting Supine and Standing Blood Pressure ^1^.

	Normal Diet (12/12)			Time-Restricted Eating (16/8)			
	Pre-	Post-	Change	Pre-	Post-	Change	*p* Value
Supine							
Systolic	113.9 ± 5.9	116.9 ± 7.5	3.0 ± 6.7	115.0 ± 8.5	116.1 ± 5.9	1.1 ± 6.4	0.58
Diastolic	74.6 ± 3.7	77.0 ± 5.6	2.4 ± 7	77.4 ± 4.9	76.6 ± 5.7	−0.9 ± 5.5	0.40
Standing							
Systolic	117.6 ± 5.2	118.2 ± 6.3	0.6 ± 5	119.7 ± 7.6	120.6 ± 5.5	0.9 ± 6.5	0.93
Diastolic	78.4 ± 3	78.2 ± 5.6	−0.2 ± 5.4	80.4 ± 4.7	79.7 ± 2.9	−0.7 ± 4.1	0.52

^1^ Values are mean ± SD for 15 subjects and units are mmHg. *p* values represent difference in the change values between diet interventions.

**Table 8 nutrients-15-00985-t008:** Cardiovascular Disease Risk Markers: Lipids and Lipoproteins ^1^.

	Normal Diet (12/12)			Time-Restricted Eating (16/8)			
	Pre-	Post-	Change	Pre-	Post-	Change	*p* Value
Total cholesterol	170.5 ± 27.4	172.1 ± 27.9	1.6 ± 15.9	177.0 ± 25.4	167.1 ± 32.9	−2.9 ± 17.8	0.55
Total triglycerides	74.1 ± 21.0	82.6 ± 51.7	8.5 ± 40.4	72.9 ± 23.7	70.5 ± 24.7	−3.0 ± 16.2	0.71
VLDLc	14.9 ± 4.2	17.0 ± 10.1	2.1 ± 8.1	15.0 ± 4.9	14.5 ± 5.0	−0.6 ± 3.3	0.46
LDLc	92.5 ± 19.9	93.8 ± 19.9	1.3 ± 10.3	99.2 ± 18.9	91.3 ± 23.6	−2.6 ± 12.1	0.31
HDLc	63.0 ± 10.4	62.2 ± 12.0	−0.8 ± 6.6	62.9 ± 10.8	61.3 ± 10.0	−0.4 ± 8.0	0.60
nHDLc	107.5 ± 20.6	109.9 ± 23.5	2.4 ± 13.0	114.2 ± 19.9	105.9 ± 26.1	−3.1 ± 13.3	0.30
TC/HDL	2.7 ± 0.3	2.8 ± 0.5	0.1 ± 0.3	2.9 ± 0.4	2.8 ± 0.4	−0.1 ± 0.3	0.06
LDL/HDL	1.5 ± 0.3	1.5 ± 0.4	−0.1 ± 0.5	1.6 ± 0.3	1.5 ± 2.9	−0.5 ± 1.6	0.65

^1^ Values are mean ± SD for 15 subjects. *p* values are for comparison of change values. All lipid units are mg/dL. Abbreviations: Very low-density lipoprotein cholesterol, VLDLc; low-density lipoprotein cholesterol, LDLc; high-density lipoprotein cholesterol, HDLc; non-HDL-cholesterol, nHDLc; ratio of total cholesterol/high-density lipoprotein, TC/HDL; ratio of low-density lipoprotein/high-density lipoprotein LDL/HDL.

## Data Availability

The data presented in this study are available on request from the corresponding author. The data are not publicly available due to privacy concerns.

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
