# Peer review of "An Intervention of Four Weeks of Time-Restricted Eating (16/8) in Male Long-Distance Runners Does Not Affect Cardiometabolic Risk Factors"

_nutrients, 2023, doi:10.3390/nu15040985_

Round 1
Reviewer 1 Report
This is a well-written and interesting manuscript on an important topic that furthers our understanding of TRE, particularly in a sports science context. I believe this manuscript will be of great interest to the readers of this special issue, as well as athletes who are interested in TRE.
I have several recommendations to strengthen the manuscript:
Introduction:
· Good, succinct, well-written introduction. My recommendation is to include in the first paragraph, a bit more context on TRE including the distinction that, unlike many other forms of IF, in TRE there is no suggested change to the content/amount of food eaten during the shortened eating window. I also think the first paragraph would benefit from a sentence or two describing the reasons for TRE and why it is proposed as a strategy for health and wellbeing (i.e. some discussion of an extended fasting period and improving digestive and metabolic processes).
· Page 5, line 80 of the intro. The authors state “a normal eating window of 12h fasting and 12h eating” – this needs some justification and context on what a ‘normal’ population is that this eating timing comes from. This may be normal for athletes, and so makes sense in this sentence, however in other populations used in TRE research typical eating windows can extend to 14-16h prior to TRE.
Methods:
· Could the authors clarify whether participants were instructed to keep their usual eating habits from prior to enrolling in the study and during the washout period? There can be a tendency for participants to automatically alter eating patterns when starting a study with a diet manipulation even if it is the timing that is the intervention.
· Was caffeine intake limited before the blood pressure recording? Fasting was mentioned so I am assuming this also meant no black coffee/tea
Results:
· Overall, results are clear and well organised
· Could average energy intake and macronutrient breakdown in the two arms be reported, and was there a difference in this intake or were participants able to keep their intake consistent?
· Did you collect any information on the timing of when participants chose to have their eating and fasting periods? This would also be useful control data to report (or a range of dates) to understand if there were differences between the participants in when they ate. I understand this information may have been in the original report, so perhaps a summary of this data and the energy intake data could be included for context for the current manuscript and results.
Discussion:
· Some discussion of TRE research in non-athlete samples would strengthen the discussion and provide context for the results. For example in the paragraph starting on line 319, the paragraph on insulin sensitivity and potentially little room for improvement in athletes, a comparison to non-athlete samples (particularly high-risk samples such as obese populations) where there are differences found in insulin sensitivity from TRE would be useful.
Author Response
Dear reviewer,
Thank you for reviewing the manuscript “An Intervention of Four Weeks of Time Restricting Eating (16/8) in Male Long-Distance Runners Does Not Affect Cardiometabolic Risk Factors”. We understand that reviewing a manuscript can require extensive time and attention to detail, so we greatly appreciate your efforts in making our manuscript better and for your insightful recommendations. Please see below our responses to your comments.
Academic Editor Comments:
Not having measured performance, with for example a simple time to exhaustion test at 5 km race pace or similar, is a major weakness of the study. I spotted a few minor errors already in the Abstract for example kg units for percent body fat, so the manuscript should be checked in detail for errors.
Response: We acknowledge that an exercise performance measure should be included. We addressed that in our companion paper “Four Weeks of 16/8 Time Restrictive Feeding in Endurance Trained Male Runners Decreases Fat Mass, without Affecting Exercise Performance”. We added a brief description of our performance results and a reference on lines 52-56 and 284. The error in units for body fat percentage was corrected in the abstract on line 26.
Reviewer 1.
My recommendation is to include in the first paragraph, a bit more context on TRE including the distinction that, unlike many other forms of IF, in TRE there is no suggested change to the content/amount of food eaten during the shortened eating window. I also think the first paragraph would benefit from a sentence or two describing the reasons for TRE and why it is proposed as a strategy for health and wellbeing (i.e. some discussion of an extended fasting period and improving digestive and metabolic processes).
Response: We edited the introduction per your recommendations. The changes are highlighted in the revised manuscript on lines 39-48.
Page 5, line 80 of the intro. The authors state “a normal eating window of 12h fasting and 12h eating” – this needs some justification and context on what a ‘normal’ population is that this eating timing comes from.
Response: We added justification for using a 12-hour window for our normal feeding window per your recommendations on line 102-105.
Could the authors clarify whether participants were instructed to keep their usual eating habits from prior to enrolling in the study and during the washout period?
Response: We added clarification regarding dietary pattern during washout and prior to visit 1 on line 115-119.
Was caffeine intake limited before the blood pressure recording? Fasting was mentioned so I am assuming this also meant no black coffee/tea
Response: We added language regarding caffeine intake prior to test day on line 166-167.
Could average energy intake and macronutrient breakdown in the two arms be reported, and was there a difference in this intake or were participants able to keep their intake consistent?
Response: We added text and a table (Table 2) outlining macronutrient breakdown on line 243-247.
Did you collect any information on the timing of when participants chose to have their eating and fasting periods?
Response: A timing of eating paragraph was added on lines 239-241.
Some discussion of TRE research in non-athlete samples would strengthen the discussion and provide context for the results. For example in the paragraph starting on line 319, the paragraph on insulin sensitivity and potentially little room for improvement in athletes, a comparison to non-athlete samples (particularly high-risk samples such as obese populations) where there are differences found in insulin sensitivity from TRE would be useful.
Response: The discussion has been edited per your recommendations on lines 340-346.

Reviewer 2 Report
Comments to the Author:
I thank to the editors for the opportunity to review this study, beside I would also like to congratulate the authors for the made effort in their study. The present manuscript by Richardson et al., analyzed “An Intervention of Four Weeks of Time Restricted Eating (16/8) in Male Long-Distance Runners Does Not Affect Cardiometabolic Risk Factors”. The authors examined the effects of four weeks of the 16/8 TRE diet compared to a normal eating window of 12 h fasting and 12 h eating on body composition, resting energy expenditure, and biomarkers of cardiometabolic disease risk using a randomized crossover study design in competitive male endurance runners. The currently the paper needs a lot of information and clarification of many doubts in the experimental design.
1. The authors comment in the introduction that this type of diet is used because endurance athletes improve their athletic performance. Could you add what are these improvements with bibliographic references?. Line 64 needs a reference.
2. The authors of this study have finally used a number of 15 active individuals, however they had intended to use 27. And my question is related to how did the authors know that by dropping to 15 subjects the statistical power of the study was still sufficient? I recommend that you use the G-Power program; it will be very helpful and will give you enough participants for your study.
3. Why was the washing period not specific but with an interval of 2-4 weeks? Are you sure that the subjects who had a 2-week washout period could have affected the study?
4. From my point of view, it should have been contracted that all subjects would have had the same training schedule, I mean, either train in the morning or train in the afternoon, in this way the feeding window would have always been the same and this bias is eliminated from the experiment. Therefore, I would like to know how the authors know that these changes in training schedules have not affected the results? I mean, of the 15 subjects that you have used, how many have trained in the morning or in the afternoon and if this could have affected the study.
5. Did the authors follow the Declaration of Helsinki in their experiment?
6. In their study, the authors say that athletes must have a maximal oxygen consumption (VO2 max) ≥ 40 ml/kg/min, and be weight stable for the past 6 mo. ¿When was it analyzed it? I cannot find in the study any section in the methodology where the test that was carried out to measure the maximum oxygen consumption is explained.
7. Why was this performed? Subjects were asked to stand for 5 minutes before standing blood pressure was collected.
8. Line 225: time-restricted eating change to TRE.
9. The authors should put more effort into the discussion section and focus on discussing their results with the available literature. Also, there is too much information that makes the reader get lost. Try to drastically reduce this session and as I said focus on discussing your results, because there is no need to comment again because you have already done in the results section.

Author Response
Dear reviewer,
Thank you for reviewing the manuscript “An Intervention of Four Weeks of Time Restricting Eating (16/8) in Male Long-Distance Runners Does Not Affect Cardiometabolic Risk Factors”. We understand that reviewing a manuscript can require extensive time and attention to detail, so we greatly appreciate your efforts in making our manuscript better and for your insightful recommendations. Please see below our responses to your comments.
Academic Editor Comments:
Not having measured performance, with for example a simple time to exhaustion test at 5 km race pace or similar, is a major weakness of the study. I spotted a few minor errors already in the Abstract for example kg units for percent body fat, so the manuscript should be checked in detail for errors.
Response: We acknowledge that an exercise performance measure should be included. We addressed that in our companion paper “Four Weeks of 16/8 Time Restrictive Feeding in Endurance Trained Male Runners Decreases Fat Mass, without Affecting Exercise Performance”. We added a brief description of our performance results and a reference on lines 52-56 and 284. The error in units for body fat percentage was corrected in the abstract on line 26.
Reviewer 2:
The authors comment in the introduction that this type of diet is used because endurance athletes improve their athletic performance. Could you add what are these improvements with bibliographic references?. Line 64 needs a reference.
Response: The statement made in the introduction is not that there is scientific evidence that TRE can improve athletic performance, but instead that athletes believe it can without such evidence. We have references included on line 48. A reference has been added for the statement that athletes are at a decreased risk for developing CVD on line 76.
The authors of this study have finally used a number of 15 active individuals, however they had intended to use 27. And my question is related to how did the authors know that by dropping to 15 subjects the statistical power of the study was still sufficient? I recommend that you use the G-Power program; it will be very helpful and will give you enough participants for your study.
Response: Our intent was not to use 27 subjects. Twenty-seven subjects were enrolled in the study total, including those that dropped out. According to our power calculation, 16 subjects were needed. We added more detail of the power analysis on lines 226-228 and the limitations of not reaching this number on lines 365-370.
Why was the washing period not specific but with an interval of 2-4 weeks? Are you sure that the subjects who had a 2-week washout period could have affected the study?
Response: We added a sentence to address this on lines 111-113 and on lines 363-365.
From my point of view, it should have been contracted that all subjects would have had the same training schedule, I mean, either train in the morning or train in the afternoon, in this way the feeding window would have always been the same and this bias is eliminated from the experiment. Therefore, I would like to know how the authors know that these changes in training schedules have not affected the results? I mean, of the 15 subjects that you have used, how many have trained in the morning or in the afternoon and if this could have affected the study.
Response: Thank you for your point and this was a detail we gave a lot of thought to. We allowed the subjects to choose the time of their training schedule for a few reasons including: 1) convenient for the subject to help with retention 2) The aim to maintain current circadian rhythm. If we had asked all subjects to train in the morning, this would have required some individuals to wake up earlier than normal to train, therefore disrupting their circadian rhythm, which also may have skewed results, particularly during their first visit before they adapt. 3) Each subject participated in both arms of the study so any bias should have been eliminated. We believe we made the best decision given the parameters listed above. We have addressed this limitation on lines 360-363.
Did the authors follow the Declaration of Helsinki in their experiment?
Response: Yes, the declaration of Helsinki was followed as stated in lines 122-124
In their study, the authors say that athletes must have a maximal oxygen consumption (VO2 max) ≥ 40 ml/kg/min, and be weight stable for the past 6 mo. When was it analyzed it? I cannot find in the study any section in the methodology where the test that was carried out to measure the maximum oxygen consumption is explained.
Response: VO2 max was determined during the familiarization visit and details can be found in the paragraph entitled “familiarization visit” that we added on lines 138-162. Regarding subjects being weight stable for 6 months-this was self-reported.
Why was this performed? Subjects were asked to stand for 5 minutes before standing blood pressure was collected.
Response: Low energy availability due to intermittent fasting could possibly cause orthostatic intolerance due to reduced sympathetic circulatory responses. We added this statement on line 69-70 and added references. We also added the reference for detecting orthostatic intolerance as measured by examining the difference in supine and standing 5 min blood pressure where blood pressure should increase to compensate for the effects of gravity on line 187-189.
Line 225: time-restricted eating change to TRE
Response: Thank you for pointing this out. We have changed this on line 287
The authors should put more effort into the discussion section and focus on discussing their results with the available literature. Also, there is too much information that makes the reader get lost. Try to drastically reduce this session and as I said focus on discussing your results, because there is no need to comment again because you have already done in the results section.
Response: We appreciate your comments and have reduced the discussion to focus just on our results compared to other studies. Please see our shortened discussion section.
